# GAUGE-INVARIANT REPRESENTATION HOLONOMY

**Vasileios Sevetlidis,**[*] **& George Pavlidis**
Athena Research Center
Kimmeria Campus
Xanthi, GR-67100, Greece
{vasiseve,gpavlid}@athenarc.gr

## ABSTRACT

Deep networks learn internal representations whose geometry—how features bend, rotate, and evolve—affects both generalization and robustness. Existing similarity measures such as CKA or SVCCA capture pointwise overlap between activation sets, but miss how representations change along input paths. Two models may appear nearly identical under these metrics yet respond very differently to perturbations or adversarial stress. We introduce representation holonomy, a gauge-invariant statistic that measures this path dependence. Conceptually, holonomy quantifies the "twist" accumulated when features are parallel-transported around a small loop in input space: flat representations yield zero holonomy, while nonzero values reveal hidden curvature. Our estimator fixes gauge through global whitening, aligns neighborhoods using shared subspaces and rotation-only Procrustes, and embeds the result back to the full feature space. We prove invariance to orthogonal (and affine, post-whitening) transformations, establish a linear null for affine layers, and show that holonomy vanishes at small radii. Empirically, holonomy increases with loop radius, separates models that appear similar under CKA, and correlates with adversarial and corruption robustness. It also tracks training dynamics as features form and stabilize. Together, these results position representation holonomy as a practical and scalable diagnostic for probing the geometric structure of learned representations beyond pointwise similarity.

## 1 INTRODUCTION

Modern deep networks learn internal representations whose geometry—how features orient, align, and evolve—matters for generalization and robustness. Yet most standard diagnostics are *pointwise*: they compare two activation sets on a fixed dataset using singular vector canonical correlation analysis (SVCCA), projection-weighted CCA (PWCCA), centered kernel alignment (CKA), or representational similarity analysis (RSA) thereby judging subspace overlap while remaining blind to how features *move* as inputs are varied along natural directions (pose, illumination, texture) (Raghu et al., 2017; Morcos et al., 2018; Kornblith et al., 2019; Kriegeskorte et al., 2008). This leaves a practical gap: two models can appear highly similar under CKA or CCA, and still behave differently under adversarial or corruption stress because their intermediate features rotate differently along input paths.

We address this gap by turning alignment itself into an object of study. We view a layer's representation as a field over input (or transformation) space and endow it with a *discrete connection*: between nearby inputs we estimate a shared principal subspace and compute the optimal special-orthogonal alignment (rotation-only Procrustes) of the two local feature clouds; composing these small rotations around a closed loop yields a single orthogonal matrix whose deviation from identity we call *representation holonomy*. Nonzero holonomy indicates path-dependent (nonintegrable) transport in the classical sense of connections and their curvature (Ambrose and Singer, 1953). The construction is *gauge-invariant* by design: global whitening fixes a sensible gauge by removing second-order anisotropy; orthogonal reparameterizations of layers leave the statistic unchanged; restricting to a low-rank shared subspace improves stability and cost (Schönemann, 1966; Kabsch, 1976; Kessy et al., 2018; Björck and Golub, 1973; Davis and Kahan, 1970).

---

[*]Corresponding author: VS, ORCID VS: 0000-0001-9348-8786, GP: 0000-0002-9909-1584

Our proposal complements two nearby lines of work rather than competing with them. First, local equivariance tests (e.g., Lie-derivative "local equivariance error") quantify *infinitesimal* sensitivity but do not assess global path-dependence via loop composition (Lenc and Vedaldi, 2015; Gruver et al., 2022). Second, gauge-/manifold-equivariant architectures build a connection into the model so that desired transports are integrable by design; we instead *measure* the emergent transport of standard models, providing a diagnostic that travels with existing practice in vision architectures (Bronstein et al., 2021; Cohen et al., 2019; Schonsheck et al., 2018; Masci et al., 2015). In downstream terms, holonomy gives a compact, layer-wise summary of pathwise geometry that (i) is inexpensive to compute, (ii) scales to common backbones, and (iii) adds information orthogonal to pointwise similarity, making it a natural candidate to relate feature geometry to robustness (Hendrycks and Dietterich, 2019).

**Contributions.** (1) We propose a practical estimator of *representation holonomy* that combines global whitening, shared-neighbor subspaces, and rotation-only Procrustes alignment. The estimator is explicitly gauge-invariant and stable in the small-radius limit. (2) We prove formal invariances (orthogonal and, after whitening, affine), establish a *linear null* showing affine layers yield zero holonomy, and derive a *small-radius limit* where holonomy vanishes linearly with loop radius. A perturbation analysis (Procrustes + Davis–Kahan/Wedin) provides explicit finite-sample and truncation error bounds (Schönemann, 1966; Björck and Golub, 1973; Davis and Kahan, 1970; Kessy et al., 2018). (3) On MNIST/MLP and CIFAR-10/100 with ResNet-18, we show that holonomy (i) increases with loop radius and depth even when CKA remains high, revealing pathwise geometry beyond pointwise similarity; (ii) rises during training as features form and stabilizes at convergence; and (iii) correlates with adversarial and corruption robustness across training regimes including ERM, label smoothing, mixup, and adversarial training (Kornblith et al., 2019; Hendrycks and Dietterich, 2019).

Section 2 situates our work among similarity metrics, equivariance diagnostics, and gauge-/manifold-equivariant architectures. Section 3 formalizes the discrete connection and holonomy estimator and establishes invariance and small-radius results with finite-sample error bounds. Section 4 reports controlled loops, training dynamics, robustness studies, and ablations (whitening choice, SO vs. O, neighbor sharing, and $k/q$ sensitivity). Section 6 summarizes limitations and implications, and we release code and seeded configs for full reproducibility.

## 2 RELATED WORK

Comparing learned representations across networks and inputs is complicated by the fact that layer activations admit many equivalent parameterizations, i.e., a gauge freedom that allows local changes of basis without altering function. A large body of work therefore develops basis-invariant or basis-robust comparison tools. CCA-based approaches—SVCCA and PWCCA—compare the subspaces spanned by activations across models or training checkpoints, reducing sensitivity to neuron permutations while still depending on preprocessing choices and data coverage (Raghu et al., 2017; Morcos et al., 2018). Kernel-based Centered Kernel Alignment (linear and nonlinear CKA) has emerged as a simple and reliable alternative with improved stability across architectures, layers, and seeds, and clear links to representational similarity analysis (RSA) from systems neuroscience (Kornblith et al., 2019; Kriegeskorte et al., 2008). Beyond scalar similarities, a complementary line aligns entire representations by explicit linear transports: orthogonal Procrustes (and its $\det = +1$ Kabsch variant) yields optimal rotation-only maps between paired activation matrices, while principal angles quantify shared subspaces; classical perturbation theory (Davis–Kahan/Wedin) provides finite-sample error control for the estimated subspaces and transports (Schönemann, 1966; Kabsch, 1976; Björck and Golub, 1973; Davis and Kahan, 1970). Preprocessing is itself a gauge choice: statistically principled whitening schemes such as ZCA-corr justify a fixed global gauge that removes second-order anisotropy before any local alignment (Kessy et al., 2018). Parallel to these comparison methods, empirical tests of (approximate) equivariance and model equivalence probe how features change under controlled input transformations; early work proposed finite-difference tests for CNNs, and more recent formulations use Lie derivatives to define a *local equivariance error* that is differential and layer-wise (Lenc and Vedaldi, 2015; Gruver et al., 2022). Geometric deep learning places these observations in a coordinate-free framework: data domains (e.g., manifolds) come with local frames (gauges), and operations should be gauge-aware; gauge-equivariant CNNs make the connection (in the differential-geometric sense) an architectural primitive, while manifold convolutions based on

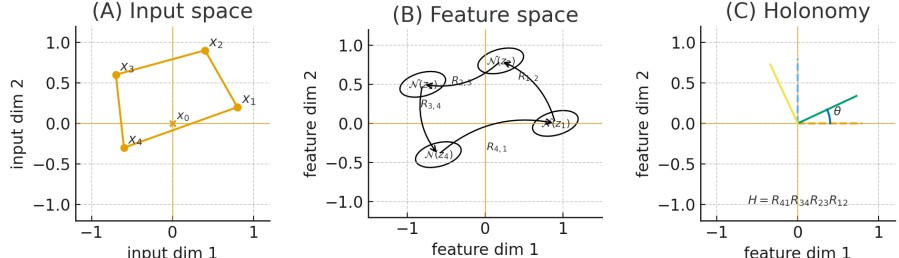

Figure 1: **Holonomy as path-dependent feature rotation.** (A) A small closed loop $\gamma = (x_0, \ldots, x_{L-1}, x_L{=}x_0)$ in a 2D input slice. (B) The corresponding features $z_i = z(x_i)$ and their local neighbourhoods $\mathcal{N}(z_i)$; for each edge we estimate an orthogonal transport $R_{i,i+1}$ that best aligns the two nearby feature clouds. (C) Composing these transports around the loop yields the holonomy $H = R_{L-1} \cdots R_1 R_0$, visualised as the net rotation of a reference direction by angle $\theta$. Holonomy is invariant to layer-wise gauge changes (global change of feature basis) and measures how much the representation "twists" when inputs follow a loop, rather than just how similar activations are at individual points.

parallel transport or geodesic patches move features intrinsically across space (Bronstein et al., 2021; Cohen et al., 2019; Schonsheck et al., 2018; Masci et al., 2015). Relatedly, Riemannian approaches on structured spaces (e.g., SPD/Grassmann) deploy intrinsic means, transports, and normalizations inside networks, underscoring the usefulness of connection-like operations on representation manifolds (Huang and Van Gool, 2017; Brooks et al., 2019). Finally, robustness benchmarks such as ImageNet-C/P offer downstream behavioral checks; because they include *sequences* of small perturbations, they are natural testbeds for path-sensitive phenomena in representation geometry (Hendrycks and Dietterich, 2019).

*This paper* adopts the geometric viewpoint but applies it *as a diagnostic* to standard models rather than as an architectural constraint. We model layer-wise representations over data (or transformation) space as sections of a vector bundle and make the alignment rule itself a *connection*: locally, we estimate a shared subspace (with principled whitening as a fixed gauge) and define the transport between nearby inputs by the optimal special-orthogonal map in that subspace; globally, we compose these local transports around closed loops and quantify the resulting *holonomy*. By construction, our measurement is invariant to per-layer orthogonal reparameterizations (gauge transforms) and robust to admissible whitening choices, distinguishing it from scalar, path-agnostic similarities such as CKA/RSA and from single-step Procrustes alignments (Kornblith et al., 2019; Kriegeskorte et al., 2008; Schönemann, 1966). The Ambrose–Singer perspective links small-loop holonomy to curvature, yielding concrete predictions that we test empirically; in particular, we show that networks can be locally near-equivariant (small Lie-derivative error) yet exhibit nontrivial *global* holonomy that correlates with stability under perturbation sequences, a phenomenon invisible to standard similarity scores (Ambrose and Singer, 1953; Gruver et al., 2022; Hendrycks and Dietterich, 2019).

## 3 REPRESENTATION HOLONOMY

Intuitively, a layer's representation assigns to each input $x$ a feature vector $z(x) \in \mathbb{R}^p$. If we move $x$ along a small closed loop $\gamma$ in input space (for example by composing small transformations), the corresponding features $z(x)$ trace out a loop in representation space. Locally, between two nearby points on the loop we can align their feature neighbourhoods by an orthogonal map $R_{i,i+1} \in \mathrm{SO}(p)$ that best matches the two clouds (Figure 1, panels A–B). Composing these local transports around the entire loop yields a net rotation $H = R_{L-1} \cdots R_1 R_0$ (panel C). If the representation were perfectly "flat" along $\gamma$—for instance, if it were globally linear and we controlled for gauge—this product would be the identity. Deviations of $H$ from $I$ therefore quantify the path dependence (curvature) of the learned features, and are insensitive to global changes of feature basis.

At a high level, the proofs rely on three standard tools: (i) Procrustes alignment in shared low-dimensional subspaces, (ii) matrix perturbation bounds of Davis–Kahan/Wedin type for controlling subspace errors, and (iii) finite-sample concentration bounds for covariance and whitening operators.

We collect the technical details in Appendix S.3–S.5. At a given layer, we call a transform $z(x) \mapsto Qz(x) + b$, with $Q \in O(p)$ (after whitening) and $b \in \mathbb{R}^p$, a *gauge transformation*: an input-independent change of basis in feature space. Two networks related by such transforms at internal layers are representation-equivalent. Our estimator is invariant to these transformations (and, after whitening, to general invertible affine reparameterisations), so that holonomy reflects only relative orientation changes induced by input paths rather than arbitrary basis choices.

Let $f : \mathbb{R}^d \to \mathbb{R}^C$ be a classifier and let $z : \mathbb{R}^d \to \mathbb{R}^p$ denote a fixed layer's representation (layer index suppressed). For inputs $x \in \mathcal{X} \subset \mathbb{R}^d$ we write $z(x) \in \mathbb{R}^p$. Given a small loop $\gamma = (x_0, \ldots, x_{L-1}, x_L{=}x_0)$ in input (or transformation) space, we define a local linear transport $R_i \in \mathrm{SO}(p)$ between the features at successive points and take the *holonomy*

$$H(\gamma) \;=\; R_{L-1} \cdots R_1\, R_0 \;\in\; \mathrm{SO}(p), \qquad h_{\mathrm{norm}}(\gamma) \;=\; \frac{\|H(\gamma) - I\|_F}{2\sqrt{p}} \;\in\; [0,1], \qquad (1)$$

reporting also the eigen–angle multiset $\{\theta_j\}_{j=1}^p$ of $H(\gamma)$ (eigenvalues $e^{i\theta_j}$ on the unit circle). Conceptually, if $z$ is $C^2$ then first-order linearization suggests $R_i \;=\; I + \mathrm{O}c(\|x_{i+1} - x_i\|)$, hence $H(\gamma) = I + \mathrm{O}c(\mathrm{length}(\gamma))$.

**Estimator (used in practice).** We pool a set $\mathcal{N}$ of examples, compute features $Z = \{z(x)\}_{x \in \mathcal{N}}$, their mean $\mu$ and covariance $\Sigma$, and fix a global gauge by *whitening* $\tilde{z}(x) = \Sigma^{-1/2}(z(x) - \mu)$ (ZCA-corr; any fixed symmetric square root suffices) (Kessy et al., 2018). For an edge $(x_i, x_{i+1})$, let $m_i = \frac{1}{2}(\tilde{z}(x_i) + \tilde{z}(x_{i+1}))$ and choose a *shared* index set $\mathcal{I}_i$ of size $k$ as the $k$-NN of $m_i$ in the whitened pool. On these same rows we compute a *shared* soft center at the midpoint $\bar{\mu}_i = \sum_{j \in I_i} w_j^{(i)} \tilde{Z}_{j:}$ with weights $w_j^{(i)} \propto \exp(-\|\tilde{Z}_{j:} - m_i\|/\sigma_i)$, and set $X_i = Y_i = \tilde{Z}_{I_i} - \bar{\mu}_i$. Let $W_i = \mathrm{diag}(w_j^{(i)})_{j \in I_i}$ and $B_i \in \mathbb{R}^{p \times q}$ be the top-$q$ right singular vectors of $\begin{bmatrix} X_i \\ Y_i \end{bmatrix}$. In $\mathbb{R}^q$ we solve orthogonal Procrustes: if $U_i \Sigma_i V_i^\top = \mathrm{SVD}((X_i B_i)^\top W_i (Y_i B_i))$ then $R_i^{(q)} = U_i V_i^\top \in \mathrm{SO}(q)$ (enforce $\det = +1$) (Schönemann, 1966; Kabsch, 1976). We *embed* back to $\mathbb{R}^p$ by

$$\widehat{R}_i \;=\; B_i R_i^{(q)} B_i^\top + (I - B_i B_i^\top) \;\in\; \mathrm{SO}(p), \qquad (2)$$

compose $\widehat{H}(\gamma) = \widehat{R}_{L-1} \cdots \widehat{R}_0$, and report $\widehat{h}_{\mathrm{norm}} = \|\widehat{H} - I\|_F/(2\sqrt{p})$ together with eigen–angles of $\widehat{H}$. Indeed, $(I - BB^\top)B = 0$ so $\widehat{R}_i^\top \widehat{R}_i = I$; moreover $\det \widehat{R}_i = \det R_i^{(q)} = +1$. This construction is inexpensive (small SVDs in a shared subspace) and numerically stable.

**Structural properties (statements; full proofs in App. S.1).** [1] (i) *Gauge invariance.* If whitened features are reparameterized by any $U \in O(p)$, i.e., $\tilde{z}'(x) = U\tilde{z}(x)$, then the shared indices $\mathcal{I}_i$ are unchanged, $\widehat{R}_i' = U \widehat{R}_i U^\top$, and $\widehat{H}' = U \widehat{H} U^\top$. Thus $\|\widehat{H}' - I\|_F = \|\widehat{H} - I\|_F$ and the eigen–angle multiset is identical. (ii) *Affine invariance (post-whitening).* For any invertible affine map on raw features, $z'(x) = Az(x) + b$, whitening by the corresponding pool statistics yields $\tilde{z}'(x) = Q\tilde{z}(x)$ with $Q \in O(p)$ (because $\Sigma'^{-1/2} A \Sigma^{1/2}$ is orthogonal when $\Sigma' = A\Sigma A^\top$), hence the previous item applies. (iii) *Linear null.* If $z(x) = Bx + c$ is affine and each edge uses shared rows, then $X_i = Y_i$ for all $i$ and $\widehat{R}_i = I$, so $\widehat{H}(\gamma) = I$. (iv) *Orientation/cycling.* Reversing a loop inverts holonomy, $\widehat{H}(\gamma^{-1}) = \widehat{H}(\gamma)^{-1}$, so the Frobenius gap is unchanged; cyclic reparameterizations of $\gamma$ leave $\widehat{H}$ unchanged. (v) *Normalization.* For any $H \in O(p)$ with eigen–angles $\{\theta_j\}$, $\|H - I\|_F^2 = 2\sum_{j=1}^p (1 - \cos\theta_j) \le 4p$, hence $h_{\mathrm{norm}} \in [0,1]$ with equality 1 iff all $\theta_j = \pi$. All invariance statements apply to the post-readout features; non-invertible readouts (e.g., JL) are outside the affine-invariance claim.

**Small-radius behavior (statement; proof in App. S.2).** Assume $z$ is $C^2$ with Lipschitz Jacobian on a neighborhood of $\gamma_r$, the loop $\gamma_r$ has total length $\mathrm{O}c(r)$, the shared-midpoint $k$-NN has overlap

---

[1] **Pointers to supplement**: App. S.0 fixes notation; App. S.1 gives full proofs of invariances, nulls, and normalization; App. S.2 proves the small-radius limit; App. S.3 states a Procrustes perturbation lemma; App. S.4 handles subspace truncation; App. S.5 derives per-edge and holonomy error bounds; App. S.6 provides an explicit algorithm and App. S.7–S.8 cover complexity and practical implications.

probability $1 - Oc(r)$ as $r \to 0$, and the subspace rank $q$ covers the local feature rank. Then for each edge $\|\widehat{R}_i - I\|_F = Oc(r)$ and

$$\|\widehat{H}(\gamma_r) - I\|_F = Oc(r), \qquad \text{hence} \quad \widehat{h}_{\text{norm}}(\gamma_r) = Oc(r). \tag{3}$$

Intuitively, shared-row centering cancels translations; Lipschitz variation of $J_z$ makes the optimal rotation deviate from $I$ by $Oc(\|x_{i+1} - x_i\|)$; products of $I + Oc(r)$ along $L = Oc(1)$ edges yield an overall $Oc(r)$ deviation.

**Estimator stability and error decomposition (statement; full derivation in App. S.5).** Under standard sampling assumptions for the neighbor pool (sub-Gaussian rows; a spectral gap $\Delta$ separating the top-$q$ right-singular subspace), the per-edge error relative to the population transport $R_i^\star$ obeys

$$\|\widehat{R}_i - R_i^\star\|_F \leq C_1 \underbrace{k^{-1/2}}_{\text{finite sample}} + C_2 \underbrace{\frac{\|\Pi_\perp^i \Sigma_i^{1/2}\|_F}{\lambda_q(\Sigma_i)^{1/2}}}_{\text{subspace truncation}} + C_3 \underbrace{\text{TV}(\mathcal{I}_i, \mathcal{I}_i^\star)}_{\text{index mismatch}} + C_4 \underbrace{\|J_z(x_{i+1}) - J_z(x_i)\|_2}_{\text{curvature}},$$

$$\tag{4}$$

with $\Pi_\perp^i = I - B_i B_i^\top$ and constants depending smoothly on local condition numbers; composing over $L = Oc(1)$ edges yields the holonomy error bound. Here $\lambda_q(\Sigma_i)$ denotes the $q$-th largest eigenvalue of the *population* covariance $\Sigma_i$ on the shared rows. The finite-sample term follows from Procrustes perturbation via singular-subspace angles, the truncation term from Davis–Kahan/Wedin, and the curvature/mismatch terms from continuity of $J_z$ and the shared-midpoint $k$-NN (Björck and Golub, 1973; Davis and Kahan, 1970).

Empirically, this decomposition matches the behaviour we observe in the vision experiments. Choosing $k$ moderately large and $q$ smaller (e.g., $k \in \{96, 128, 192\}$, $q \in \{32, 64, 96\}$ on MNIST hidden 1) keeps the finite-sample and truncation terms small: $h_{\text{norm}}$ varies only at the level of a few $10^{-7}$ across this grid, without qualitative changes. The shared-midpoint $k$-NN construction effectively controls the index-mismatch term $\text{TV}(\mathcal{I}_i, \mathcal{I}_i^\star)$: when we deliberately use disjoint neighbour sets for the two endpoints, holonomy increases markedly and becomes unstable at small radii. Finally, in linear or self-loop settings (affine networks, $r = 0$ loops), $h_{\text{norm}}$ collapses to the numerical floor ($\sim 10^{-8}$–$10^{-7}$), indicating that once finite-sample, truncation, and index-mismatch effects are controlled, the remaining signal is consistent with genuine curvature of the learned representation field.

Per edge, forming the shared $q$-subspace (thin SVD of a $(2k) \times p$ stack) costs $Oc(kpq)$, Procrustes in $\mathbb{R}^q$ costs $Oc(q^3)$, and embedding costs $Oc(pq)$; thus a loop costs $Oc(L(kpq + q^3))$, typically dominated by the subspace SVD. In practice, choose $k \gg q$ (e.g., $k \in [128, 192]$ with $q \in [64, 96]$ for vision layers), keep loop radii small to ensure neighbor overlap, use a fixed global whitening, and project to SO (not O) to avoid reflection flips (Schönemann, 1966; Kabsch, 1976; Kessy et al., 2018). Per-neighborhood whitening induces stepwise gauge drift; allowing reflections ($O(p)$) introduces $\pi$-flips; and using disjoint neighbor sets increases index noise—all create a non-vanishing bias floor as $r \to 0$. The combination of global whitening, shared neighbors, SO-only Procrustes, and subspace transport removes this bias and restores the small-radius limit.

## 4 EXPERIMENTAL PROTOCOL

We study whether representation holonomy is *valid*, *reliable*, and *useful*. This section describes datasets, models, training, readout/gauge fixing, loop construction, and the estimator. All figures use the same code and seeded configs; details (incl. exact hyperparameters and scripts to regenerate CSVs/plots) are in the Supplement.

Convolutional feature maps are globally averaged ($2 \times 2$ adaptive pooling only where explicitly stated), flattened, and projected by a fixed orthonormal Johnson–Lindenstrauss map to $p^\star = 1024$ if needed (only when the post-readout feature dimension $p^\star$ exceeds 1024). Empirically this changes $h_{\text{norm}}$ negligibly while reducing memory and runtime[2]. We fix gauge using global featurewise mean–variance standardization from a model-agnostic pool of $N_{\text{pool}} = 2048$ representations. *Note:*

---

[2]see Appendix S.B for a short numerical check

Table 1: Setup at a glance

| | |
|---|---|
| Datasets | MNIST; CIFAR-10/100 (standard splits; MNIST: $10°$ rot.; CIFAR: crop+flip) |
| Models | MNIST: 2-layer MLP (512); CIFAR: ResNet-18 ($3\times3$ stem; no max-pool) |
| Training | Adam; MNIST: 5 ep, lr $2\times10^{-3}$, wd $10^{-4}$; |
| | CIFAR-10: 8 ep, lr $10^{-3}$, wd $5\times10^{-4}$; CIFAR-10: 12 ep, lr $10^{-3}$, wd $5\times10^{-4}$ |
| Regimes (C10) | ERM; label smoothing $\varepsilon=0.1$; mixup $\alpha=0.2$; short PGD (step $2/255$, $\varepsilon=4/255$, 3 steps) |
| Readout | GAP ($2\times2$ adaptive only where noted); JL to $p^\star=1024$ if $p > p^\star$ |
| Gauge fixing | Global featurewise z-score using a model-agnostic pool $N_{\text{pool}}=2048$ |
| Loops | Per test $x_0$: 2D PCA plane from 512 nearest training neighbors (pixels); $n=12$-point circle |
| Radii | MNIST: $\{0.01, 0.02, 0.05, 0.10, 0.20\}$; CIFAR: $\{0.02, 0.05, 0.10, 0.20\}$ |
| Estimator | Shared-midpoint $k$-NN; soft centering; joint $q$-dim. subspace; $SO(q)$ Procrustes; embed to $SO(p)$ |
| Defaults | MNIST: $(k,q)=(128,64)$; CIFAR `layer2`: $(192,96)$; seeds $= 5$ |

our theory assumes full-covariance whitening; we empirically compare z-scoring vs. ZCA-corr in the Supplement and find similar outcomes in our settings. For each held-out test image $x_0$, we form a local 2D PCA plane using its 512 nearest training neighbors (in pixel space) and sample a regular $n=12$-point circle of radius $r$. Varying $n_{\text{points}} \in \{6, 8, 12, 16, 24\}$ changes $h_{\text{norm}}$ smoothly and by less than $1.2 \times 10^{-7}$, with no sign of instability[3]. We report results across the radii sets above. For each edge on the loop: (i) find a *shared* $k$-NN in whitened space at the edge midpoint; (ii) softly center both point clouds; (iii) learn a shared $q$-dimensional right-singular subspace from the stacked clouds; (iv) solve an $SO(q)$ Procrustes alignment; (v) embed back to $\mathbb{R}^p$ as an $SO(p)$ rotation. Composing edges yields $H(\gamma)$ and $h_{\text{norm}} = \frac{\|H(\gamma)-I\|_F}{2\sqrt{p}}$, with $p$ the post-readout dimension. Unless stated, defaults are as above. Reported intervals are as described in section 4 (Uncertainty and statistical reporting). Unless otherwise specified, we report two-tailed Pearson $r$ and Spearman $\rho$ computed over (regime, seed) pairs ($n = 20$). Partial correlations " | clean" residualize both variables on clean accuracy via OLS and correlate the residuals. Regression coefficients are standardized (z-scored predictors and targets); we report the coefficient $\beta$ for holonomy together with its standard error (SE), $p$-value, and adjusted $R^2$ of the full model (holonomy + clean accuracy). For small-radius behavior we fit $h_{\text{norm}} = \alpha + \beta r$ on $r \in \{0.02, 0.05, 0.10\}$ and report a nonparametric 95% bootstrap CI for $\beta$ using 4,000 resamples over the (regime, seed) rows. Error bars in small-radius plots denote standard error of the mean across (regime, seed) at each $r$.

## 5 RESULTS

We first study how holonomy scales with radius and depth, then examine its relationship to robustness across training regimes, and finally assess its stability and invariance properties. Figure 2 plots mean holonomy with 95% confidence intervals as a function of loop radius on MNIST and CIFAR-10. On MNIST, both hidden layers exhibit clear positive scaling. Fitted slopes are $1.54\times10^{-6}$ for Hidden 1 and $6.10\times10^{-6}$ for Hidden 2, with corresponding means at $r=0.10$ of $6.42\times10^{-7} \pm 5.74\times10^{-9}$ (Hidden 1) and $2.86\times10^{-6} \pm 2.64\times10^{-8}$ (Hidden 2).[4] The deeper layer consistently exhibits larger holonomy and stronger radius dependence in this setting. On CIFAR-10, `layer1` and `layer2` show very similar positive dependence on radius; Figure 2-bottom overlays regime-wise CIs for both layers. Fitted slopes remain positive for both layers (Layer 1: $2.52\times10^{-7}$; Layer 2: $3.66\times10^{-9}$), with means at $r=0.10$ of $6.01\times10^{-7}$ (Layer 1) and $4.45\times10^{-7}$ (Layer 2). Across datasets we thus consistently observe positive dependence on radius. Deeper layers often exhibit larger holonomy (e.g., MNIST, Figure 2-top), although this trend is not strictly monotone across all architectures, and on CIFAR-10 the first two layers are very close in magnitude (Figure 2-bottom).

To probe the small-radius regime more directly, we aggregate CIFAR-10 `layer2` across seeds and regimes and fit a line over $r \in \{0.02, 0.05, 0.10\}$ (Figure 3, left). The fitted slope is $1.44\times10^{-7}$ with a 95% bootstrap CI of $[-1.07\times10^{-7}, 4.22\times10^{-7}]$, consistent with near-linear behaviour and the $O(r)$ scaling predicted by Theorem 1.

On CIFAR-10 with ResNet-18 we consider four standard training recipes: (i) empirical risk minimisation (ERM) with cross-entropy loss; (ii) label smoothing (LS) with smoothing coefficient $\alpha = 0.1$;

---
[3]Experiment C (Appendix S.B)

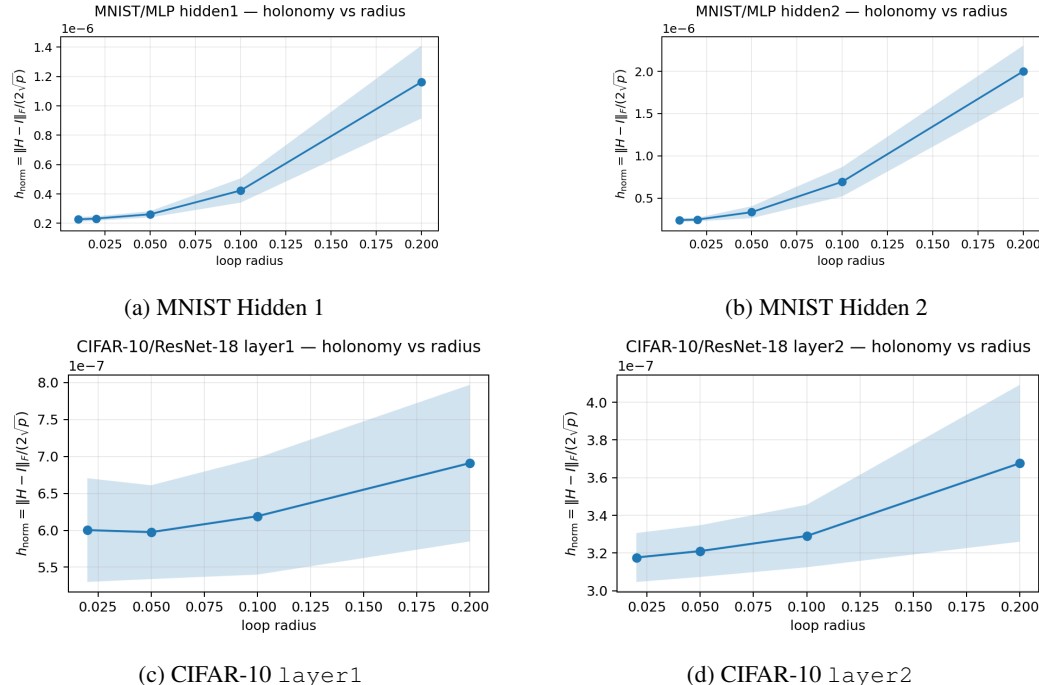

(a) MNIST Hidden 1

(b) MNIST Hidden 2

(c) CIFAR-10 `layer1`

(d) CIFAR-10 `layer2`

Figure 2: **Holonomy vs. radius on MNIST and CIFAR-10.** Mean $\pm 95\%$ CI across seeds (MNIST) and across seeds and training regimes (CIFAR-10). Both datasets exhibit positive dependence on radius; on MNIST the deeper layer has larger amplitudes, while on CIFAR-10 the first two layers are very similar in magnitude.

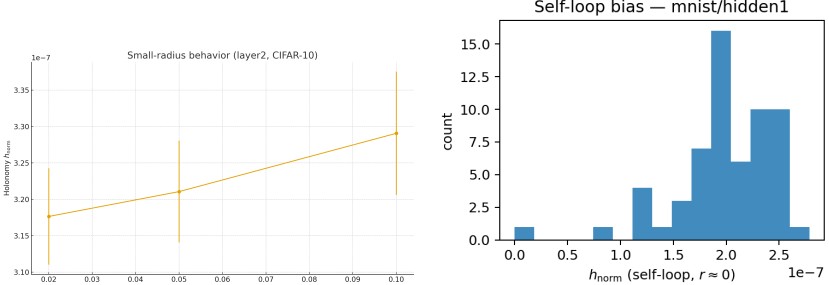

Figure 3: Small-radius regime (left) on CIFAR-10 (ResNet-18, `layer2`). Points show mean $h_{\mathrm{norm}}$ over seeds and training regimes; error bars are s.e.m. Slope estimate: $1.44 \times 10^{-7}$ (95% CI $[-1.07 \times 10^{-7}, 4.22 \times 10^{-7}]$). Self-loop bias (right) near zero (MNIST Hidden 1, $r \approx 10^{-4}$). The bias floor is $\mathcal{O}(10^{-8})$.

(iii) mixup with parameter $\alpha = 0.2$; and (iv) short projected-gradient-descent (PGD) adversarial training with $\ell_\infty$-bounded perturbations (radius $4/255$, step size $2/255$, a small number of steps). Throughout, we use "adversarial stress" to denote test accuracy under single-step FGSM and multi-step PGD-10 attacks with these hyperparameters, and "corruption stress" to denote accuracy under simple low-level corruptions (Gaussian blur, colour jitter, additive Gaussian noise), instantiated in the spirit of CIFAR-10-C-style corruptions. The robustness panel and Table 3 report clean, adversarial, and corruption accuracies for these four regimes.

At matched budgets on CIFAR-10, holonomy on `layer2` systematically varies across ERM, label smoothing, mixup, and short PGD training (Table 3). At $r = 0.10$, the adversarially trained model exhibits the largest holonomy, followed by ERM, mixup, and label smoothing. A small, single-radius slice of holonomy already associates with standard stressors: across the four regimes we

Table 2: CIFAR-10 (ResNet-18, `layer2`, radius $r = 0.1$): correlation and regression of robustness targets against holonomy $h_{\mathrm{norm}}$ with clean accuracy as control. Coefficients are standardized.

| Target | n | Pearson r | Spearman $\rho$ | Partial r \| clean | $\beta$ (std) | SE | p | Adj R² |
|---|---|---|---|---|---|---|---|---|
| fgsm acc | 20 | 0.805 | 0.565 | 0.223 | 0.080 | 0.085 | 0.36 | 0.950 |
| pgd10 | 20 | 0.809 | 0.501 | 0.276 | 0.051 | 0.043 | 0.253 | 0.987 |
| corr acc | 20 | -0.785 | -0.421 | 0.027 | 0.006 | 0.057 | 0.913 | 0.977 |

Table 3: CIFAR-10 regimes (`layer2`). Mean $h$ at $r=0.10$ and held-out accuracies from the robustness panel.

| Regime | $h_{\mathrm{norm}}$ @ $r=0.10$ | Clean Acc. (%) | FGSM Acc. (%) | Corrupt. Acc. (%) |
|---|---|---|---|---|
| ERM | $3.46\times10^{-7}$ | 82.37 | 36.54 | 57.11 |
| LabelSmooth | $3.04\times10^{-7}$ | 81.32 | 34.81 | 58.27 |
| Mixup | $3.19\times10^{-7}$ | 74.11 | 22.51 | 49.54 |
| AdvPGD | $4.74\times10^{-7}$ | 12.24 | 67.85 | 11.96 |

Correlations ($h$ vs. clean/FGSM/corrupt.): $\approx -0.96$, $\approx 0.94$, $\approx -0.96$.

observe strong correlations between mean holonomy and FGSM/corruption accuracies ($r\approx0.94$ and $r\approx-0.96$), and a corresponding inverse relation with clean accuracy ($r\approx-0.96$). Regimes that are more adversarially robust (higher FGSM accuracy) tend to have larger holonomy but lower clean and corruption accuracy, indicating that representation holonomy tracks tradeoffs along the robustness–accuracy frontier at the *regime* level.

Regime-means thus show strong descriptive correlations between holonomy and robustness across the four training recipes. However, a per-seed analysis conditioning on clean accuracy indicates only modest incremental signal: at $r = 0.10$, partial correlations are $r \approx 0.22$–$0.28$ for FGSM/PGD-10 and near zero for CIFAR-10-C (Table 2).

To isolate what holonomy adds beyond pointwise comparisons, we aligned MNIST Hidden 1 test activations with an orthogonal Procrustes map and computed linear CKA. Despite very high aligned CKA (0.987), the composed holonomy remains nonzero; the post-alignment Frobenius misfit is $2.19\times10^{-8}$, yet loop composition still accumulates a measurable twist. This control shows that near-identical pointwise representations can possess different *pathwise* geometry, and that holonomy detects those differences.

We pre-registered a sensitivity slice and ablations. At $r = 0.10$ on MNIST Hidden 1, varying $(k,q) \in \{96, 128, 192\} \times \{32, 64, 96\}$ changes $h_{\mathrm{norm}}$ by only $7.20\times10^{-7}$ end-to-end (SD $2.86\times10^{-7}$). Increasing the standardization pool from $10^3$ to $8\times10^3$ shifts $h$ by $6.49\times10^{-9}$ (from $4.05\times10^{-6}$ to $4.06\times10^{-6}$), indicating practical insensitivity to $N_{\mathrm{pool}}$. Ablations confirm that each "bias guardrail" matters: switching from $\mathrm{SO}(p)$ to $\mathrm{O}(p)$ (reflections allowed) raises $h$ by $5.37\times10^{-7}$ on average; using per-neighborhood (*local*) rather than global whitening increases $h$ by $1.59\times10^{-7}$; and, critically, dropping shared-midpoint neighbors (*separate* $k$-NNs per edge endpoint) catastrophically inflates measured holonomy (e.g., $+2.22\times10^{-1}$) even with other safeguards on. Finally, using a random plane instead of a local PCA plane reduces $h$ modestly by $1.92\times10^{-8}$ at $r=0.10$, contextualizing our loop construction choice. Varying only the loop discretisation $n_{\mathrm{points}}$ over $\{6, 8, 12, 16, 24\}$ at fixed radius on MNIST Hidden 1 yields a smooth curve with $h_{\mathrm{norm}}$ in the range $3.5$–$4.7 \times 10^{-7}$, further supporting numerical stability of the estimator with respect to loop discretisation.

A near-zero self-loop ($r\approx10^{-4}$) produces a numerically tiny bias floor on MNIST Hidden 1 (mean $4.19\times10^{-8}$; max $5.04\times10^{-8}$). A complementary small-radius study (Experiment D, Appendix S.B) on a separately trained MNIST MLP at Hidden 1 yields $h_{\mathrm{norm}} \approx 3.08 \times 10^{-7}$ for an exact self-loop ($r = 0$), and for PCA loops with radii $r \in \{10^{-3}, 2 \times 10^{-3}, 5 \times 10^{-3}, 10^{-2}, 2 \times 10^{-2}\}$ all values lie in the narrow band $h_{\mathrm{norm}} \in [2.37, 2.46] \times 10^{-7}$ (variation $\approx 3 \times 10^{-8}$). Together, these numbers characterise the numerical floor of our estimator in this setting and are consistent with the $O(r)$ small-radius behaviour predicted by Theorem 1.

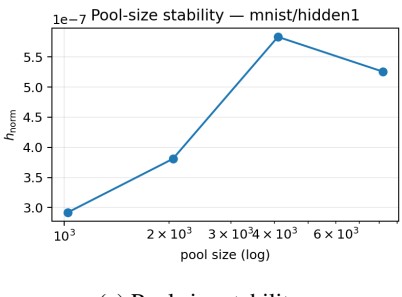

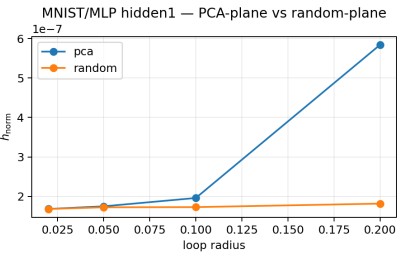

(a) Pool size stability

(b) Plane ablation (PCA vs. random)

Figure 4: **Reliability/stability.** Left: $h_{\mathrm{norm}}$ is nearly flat as $N_{\mathrm{pool}}$ increases. Right: PCA planes yield slightly higher, more geometry-aware holonomy than random planes.

Replacing nonlinearities by identity (*linear null*) drives holonomy to noise level (mean $9.57 \times 10^{-9}$; SD $2.22 \times 10^{-9}$). Gauge invariance holds: post-multiplying the readout by a random orthogonal basis changes $h$ by only $\sim 10^{-8}$ on average (MNIST: $\overline{\Delta h} = 1.17 \times 10^{-8}$; CIFAR-10: $1.65 \times 10^{-8}$) and leaves the eigen-angle spectrum near-identical (mean $L_2$ discrepancy $\approx 7.1 \times 10^{-7}$ on MNIST; $\approx 7.8 \times 10^{-7}$ on CIFAR-10). Orientation reversal behaves as expected: composing the forward loop with the inverse yields a tiny normalized gap ($7.14 \times 10^{-8}$ on MNIST; $9.53 \times 10^{-8}$ on CIFAR-10). Across datasets, layers, and training regimes, representation holonomy (i) *validly* measures a pathwise geometric effect distinct from pointwise similarity, (ii) is *reliable* under reasonable readout/estimator choices provided bias guardrails are kept, and (iii) is *useful*, describing adversarial and corruption robustness from a small, fixed-radius probe early in the network. Extended stress tests (PGD-10, CIFAR-10-C, partial correlations) and additional spectra/ablations are deferred to the Supplement.

## 6 DISCUSSION

Our estimator targets a *local, gauge-invariant* property of the learned representation field: the parallel transport induced by the network when we traverse a small input-space loop. At a given layer with feature dimension $p$, we compose per-edge transports in an estimated $q$-dimensional subspace (embedded back into $\mathrm{SO}(p)$) and summarise the loop via $h_{\mathrm{norm}} = \|H - I\|_F / (2\sqrt{p})$ and, when needed, the eigen-angle spectrum of $H$. This statistic is *complementary* to pointwise similarity measures such as CKA, SVCCA, and PWCCA: those compare unordered sets of activations at fixed inputs, while holonomy probes how features evolve along a path and whether composing local transports around a loop yields a non-trivial "twist". In particular, two networks can exhibit near-maximal aligned CKA yet differ in holonomy, indicating different pathwise geometry despite almost indistinguishable pointwise alignment; our MNIST and CIFAR-10 experiments give concrete instances of this "CKA-high but holonomy-different" regime.

Holonomy is not a single global distance between models, nor evidence of topological monodromy in data space. It captures curvature-like effects local to the family of loops under consideration, and depends on both loop design (centre, radius, plane) and the feature metric (made explicit by whitening). Our gauge choice (global whitening, shared $k$-NN at edge midpoints, rotation-only Procrustes) removes arbitrary reparameterisations of feature space, so that $h_{\mathrm{norm}}$ reflects genuine changes in representation orientation along input paths rather than artefacts of the basis.

Empirically, we find holonomy most useful in three situations. (i) *Early-epoch selection:* small-radius $h_{\mathrm{norm}}$ measured early in training already correlates with eventual robustness across regimes, providing a cheap, label-free signal for choosing runs or stopping early. (ii) *Diagnosing geometry vs. alignment:* when pointwise similarity metrics indicate that checkpoints are nearly identical, holonomy can still separate them by sensitivity to small input transports, shedding light on robustness or transfer differences that CKA alone does not explain. (iii) *Layer-wise profiling:* holonomy as a function of radius and depth highlights where the network introduces most path dependence, which can guide where to regularise or where to attach heads in transfer settings. For reliable use, our experiments suggest small radii (where $h_{\mathrm{norm}}$ scales roughly linearly), $k \gg q$ with shared-neighbour selection at midpoints, and reporting distributions (medians and IQRs) over loop centres and planes. Stability

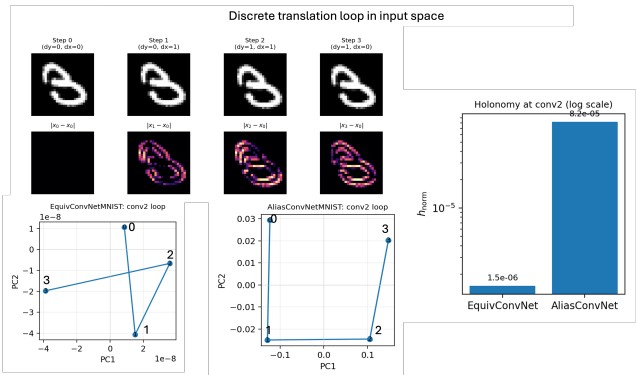

Figure 5: A single MNIST digit is translated around a small 4-step loop. At conv2, the nearly translation-equivariant CNN yields an almost closed feature loop and tiny holonomy, while the aliased CNN produces a distorted loop and holonomy about three orders of magnitude larger.

diagnostics such as neighbour-overlap IoU and the fraction of variance captured by $q$ help detect pathological settings that inflate variance.

Holonomy also has clear limitations. It is inherently *local*: it summarises curvature near the sampled loops rather than a global property of the data manifold, and results depend on how loops are constructed. PCA planes around a datum provide a reasonable default but may not always align with semantic directions, especially off-manifold. Global whitening assumes a single feature metric; strong class-conditional anisotropy can bias neighbourhoods and centres. The shared-midpoint heuristic reduces index noise but may under-represent rare modes, and rotation-only Procrustes deliberately discards scaling and shear, so scalar $h_{\mathrm{norm}}$ will under-report effects dominated by those components. Finally, although the estimator is linear-time in pool size and practical at CIFAR/ImageNet scales with compression, very deep models or dense grids of radii and planes can still be costly, so reporting confidence intervals and wall-clock helps make comparisons transparent.

Some extensions seem particularly promising. First, *beyond-local loops*: constraining loops to augmentation orbits (e.g., small rotations or translations), to domain-shift curricula, or to generative manifold paths can better align the probe with semantics and reduce off-manifold artefacts; short geodesic rectangles would directly probe commutators of input directions. Second, *richer gauges and architectures*: per-class or per-mode whitening, equivariant layers with structured gauges, and transformers with token- and position-wise gauges all offer sharper tests. Third, an especially natural application is to diffusion / score networks, where the learned score field is theoretically curl-free but in practice may deviate from this ideal; holonomy could expose such non-curl-free structure along generative trajectories.

## 7    CONCLUSION

We introduced *representation holonomy* as a gauge-invariant statistic of learned feature fields, together with a practical estimator based on shared-neighbour Procrustes transport in low-dimensional subspaces. Theoretical analysis shows that, after whitening, holonomy is invariant to affine reparameterisations, vanishes on affine maps, and scales linearly with loop radius under mild regularity assumptions, with an explicit error decomposition separating finite-sample, subspace-truncation, index-mismatch, and curvature contributions. These properties make holonomy a well-defined, local notion of "curvature" for layer-wise representations rather than an artefact of arbitrary feature bases. Holonomy is complementary to standard representation-similarity measures: networks that are almost indistinguishable under aligned CKA can still differ in holonomy and in robustness, and training recipes that change robustness also systematically modulate $h_{\mathrm{norm}}$. This supports the view of holonomy as a diagnostic tool rather than a replacement for existing metrics. Its locality and dependence on loop design make it well suited for probing specific hypotheses about representation geometry—for example, along augmentation orbits, domain-shift curricula, or generative paths—while its gauge invariance enables meaningful comparisons across checkpoints, architectures, and training regimes.

## ACKNOWLEDGEMENTS

This work has been partially supported by the ARGUS EU project (Grant Agreement No. 101132308), funded by the European Union.

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
