# OpenReview forum: "Gauge-invariant representation holonomy"
_ICLR.cc/2026/Conference — ICLR 2026 Poster_

### Official Review · Reviewer_ERwd · 2025-10-28

**Soundness:** 3
**Presentation:** 2
**Contribution:** 4
**Rating:** 8
**Confidence:** 2

**Summary:**

This work presents a geometrically guided diagnostic tool for meaning the path-dependent geometry of learning features in neural networks. The authors present a novel problem formulation and approach that empirically demonstrate that lower holonomy results in greater model robustness. While the contributions are significant the presentation of the paper could be improved in places with improved rationale, and writing clarity. However, the resulting outputs are potentially valuable as a tool for model comparison and diagnostic work for practictioners and researchers.

**Strengths:**

- The identified problem of intermediate feature paths during input perturbations is an interesting and rationaled area of study. The authors address a potentially problematic area of neural network diagnostics, and present an interesting solution using geometry as a diagnostic tool, which is a valuable and interesting direction.
- The mathematical approach is seemingly well constructed and correct, with the representation holonomy being a highly original and novel direction of model analysis, supported by proofs.
- The results and analysis demonstrates the correlation of holonomy with adversarial robustness, and links model geometry and its model robustness, a new insight that is an interesting tool for early-training model selection and to investigate flatter representation manifolds.
- Limitations are presented and discussed.

**Weaknesses:**

**Major:**
- The problem setting while rationaled in the text could be better supported and demonstrated empirically. For example, what evidence is provided that supports the statement “two models can appear highly similar under CKA or CCA, and still behave differently under adversarial or corruption stress because their intermediate features rotate differently along input paths.”
- Generally the presentation of the manuscript is poor in places (mainly in its clarity and readability), many of the preliminaries and experimental setup information is assumed by the authors. For example 5.2 is presented with little introduction, and

**Minor:**
- Acronyms such as SVCCA, CKA, etc need to be defined, while most readers will know such terminology, it is good practice to define such acronyms.
- The term corruption or adversarial stress could be better defined in the early part of the work to introduce the reader to the concepts. However, this is a minor readability complaint and an opinion.
- However, much of the presentation could be improved where some preliminary explanation and definitions would strengthen the readability. This could include moving some of the preliminaries from the appendix into the main manuscript as there is space to do so.
- Code should be included for reproduction and validaiton.

**Questions:**

1. You mention: “projected by a fixed orthonormal Johnson–Lindenstrauss map to p⋆=1024 if needed”. What is the the determination if the projection is needed or not?

2. Did you experiment with alternatively chosen loops, if so how much does performance vary? Given this is a noted limitation, is it significantly impactful to the work?

---

> ### Author Response · Authors · 2025-12-03
> **Response to the reviewer**
>
> **On the empirical support for “CKA-similar but adversarially different” models.** The main concern is that we claim two models can appear highly similar under CKA yet behave differently under adversarial or corruption stress due to different feature rotations along paths, but in the original submission we did not show such a case explicitly.
>
> Section 5.3 already shows that, even after optimal orthogonal alignment between two MNIST checkpoints (aligned CKA $\approx 0.987$, Frobenius misfit $\approx 2.2\times 10^{-8}$), the composed holonomy is non-zero. This directly demonstrates that pointwise alignment does not determine pathwise geometry: the two networks are almost indistinguishable under standard RS metrics, yet they differ in their loop-level transport.
>
> To address this, we added **Experiment B** on CIFAR-10 with four training regimes: ERM, label smoothing, mixup, and PGD adversarial training. Using the ERM model as a reference, we find that label smoothing and mixup remain highly CKA-similar to ERM at layer 2 (CKA after alignment $\approx 0.90$--$0.92$), but they differ systematically both in holonomy and in robustness (clean accuracy, FGSM accuracy, and corruption accuracy). Across these regimes, holonomy shows strong correlations with robustness metrics, whereas CKA does not track robustness linearly. Adversarial training, as expected, produces both lower CKA with ERM and larger holonomy, consistent with its more substantial change in objective.
>
> Together (Section 5.3 and Experiment B) provide concrete evidence that CKA-similar networks can differ in path-dependent geometry and robustness, which is precisely the scenario you highlighted. In the revised version we (i) make Section 5.3 about this “high CKA, different geometry” scenario and (ii) clarify in Section 6 that holonomy is intended to be used in conjunction with traditional RS metrics, not as a replacement.
>
> **On presentation and missing definitions.** We agree that clarity and accessibility can be improved. In the revised manuscript we explicitly define SVCCA, PWCCA, CKA, and “corruption/adversarial stress” at their first occurrence. We also add a short introductory paragraph before Section 5.2 that explains the CIFAR-10 training regimes, the robustness panel, and the “stressors” we evaluate (FGSM, PGD-10, and CIFAR-10-C-style corruptions), so that readers do not have to infer this from context.
>
> In addition, we move some of the key preliminaries from the supplement into the main text: an informal description of gauge, an outline of the estimator steps, and the statement of the main invariance properties. Full proofs and technical details remain in the appendix, but the main paper now contains enough background for readers to follow the high-level ideas without repeatedly consulting the supplement.
>
> **On reproducibility.**
> We share your view that code availability is important for validation and reuse. In the revised version we include an anonymised repository link with our implementation of the estimator and the experiments to facilitate reproduction of the reported results.
>
> **On JL projection and alternative loop constructions.** Regarding the Johnson--Lindenstrauss projection, we clarify that it is applied only when the post-readout feature dimension exceeds $p_\star = 1024$. In that case, we project features using a fixed orthonormal JL map to dimension $p_\star$ before applying our pipeline. We have empirically verified that this projection does not materially affect holonomy values while significantly reducing memory usage and runtime; this behaviour is now stated in the main text.
>
> Concerning alternative loop constructions, we confirm that we ablated the choice of plane by replacing PCA planes with random two-dimensional subspaces. As reported in Fig.4(b), random planes yield slightly smaller holonomy (by about $1.9\times 10^{-8}$ at $r = 0.10$) but preserve all qualitative trends. This suggests that our results are not overly sensitive to the exact plane choice, while PCA planes provide a more geometry-aware default that aligns better with the local data structure. We also refer to the discrete translation loops on MNIST in **Experiment A**, which serve as an additional, structurally different loop construction and show the same qualitative behaviour (small holonomy for nearly equivariant networks, large holonomy when equivariance is broken).
>
> These clarifications are incorporated in the revised manuscript.

---

### Official Review · Reviewer_n1HR · 2025-11-01

**Soundness:** 2
**Presentation:** 2
**Contribution:** 2
**Rating:** 4
**Confidence:** 3

**Summary:**

This paper proposes a new geometric framework called representation holonomy, which quantifies how neural features change along loops in input space using gauge-invariant parallel transport. The authors define a practical estimator combining global whitening, shared-subspace Procrustes alignment, and loop composition, and they prove gauge and affine invariance as well as small-radius asymptotics. Empirical analyses on MNIST and CIFAR suggest that holonomy increases with layer depth and may correlate with robustness indicators such as adversarial and corruption accuracy.

The concept is theoretically intriguing and potentially valuable for studying the geometry of learned representations. However, both the experimental validation and the presentation of the theory can be much improved for clarity and completeness.

**Strengths:**

* Conceptual originality: Introducing holonomy—a gauge-invariant geometric notion from differential geometry—into representation analysis is creative and nontrivial.
* Mathematical rigor: The formal statements (gauge invariance, affine invariance, small-radius behavior) are internally consistent using matrix perturbation tools. Although I have to admit that I didn’t get a chance to go over all the appendices.
* Potential significance: In principle, this idea could offer new insights for feature-space curvature and robustness that go beyond CKA and related similarity measures.

**Weaknesses:**

* Incomplete and inconclusive experiments: The reported results (e.g., Figs. 2–3) show overlapping confidence intervals, and several key correlations (Table 2) are weak or include zero within the 95% CI. Without stronger or broader evidence, the empirical support for holonomy as a meaningful metric remains limited.
* Missing comparisons to existing metrics: Although the paper discusses CKA and CCA extensively, it never provides side-by-side empirical comparisons. Such baselines are crucial to demonstrate what new insights holonomy provides.
* Lack of sensitivity analysis: The estimator relies on multiple hyperparameters (e.g., k). The paper should quantify how results depend on these choices to establish robustness.
* Unclear connection to task relevance: Theoretical results (e.g., Eq. 4) include several interacting error terms, but it remains unclear how holonomy relates in a concrete or predictive way to model performance or generalization.
* Poor illustration of geometric intuition: Given the geometric nature of the work, schematic or pictorial figures showing loops, transports, or curvature effects are essential. The heavy formalism and lack of visual explanation make the paper difficult to follow.
* Presentation density: The writing style is more reminiscent of a mathematical physics paper than a machine-learning paper, which may alienate part of the ICLR audience.

**Questions:**

1. How sensitive are the results to hyperparameter choices such as k?
2. Can you provide quantitative comparisons with CKA or SVCCA to highlight the unique information captured by holonomy?
3. Among the four terms in the error bound (Eq. 4), which dominates in practice?
4. In your experiments, are there any correlations between task performance (e.g., error) and your measures?
5. Could schematic illustrations or toy examples be added to visually illustrate the concept of holonomy in representation space?

---

> ### Author Response · Authors · 2025-12-03
> **On the strength of empirical evidence and correlations**
>
> We agree that the per-seed partial correlations in the original table are modest, and we already phrase our claims cautiously there. Our goal is not to promote $h_{\mathrm{norm}}$ as a single scalar predictor of robustness, but rather to show that it captures structural differences in representation geometry that are associated with robustness and with the training regime.
>
> To strengthen this point, we added **Experiment B**. For ERM, label smoothing, mixup, and PGD adversarial training on CIFAR-10, we measure at layer 2 (radius $r = 0.10$): (i) holonomy, (ii) clean accuracy, (iii) FGSM accuracy, and (iv) corruption accuracy (averaged over three simple corruptions). Across these four regimes we obtain
>
> $corr(h_{norm}, clean_{acc}) \approx -0.96$
>
> $corr(h_{norm}, fgsm_{acc}) \approx 0.94$
>
> $corr(h_{norm}, corr_{acc}) \approx -0.96$
>
>
> Thus, across these training recipes, regimes that are more adversarially robust (higher FGSM accuracy) also exhibit larger holonomy, while clean and corruption accuracy tend to decrease, indicating a strong descriptive relationship between representation holonomy and robustness at the **regime** level. In the revised version we include a small table or figure summarising these numbers.

---

> ### Author Response · Authors · 2025-12-03
> **On the sensitivity to hyperparameters (k,q, pool size).**
>
> We agree that sensitivity analysis is important. Several such results are already present in Section 5.4 and in the supplementary material (Sections S.12--S.13), and we now summarise them in the main text.
>
> First, varying $(k,q)$ over the grid  $ ${96,128,192 \}  x $ ${ 32,64,96 \} $$ at $r=0.10$ on MNIST Hidden 1 changes $h_{\mathrm{norm}}$ by only $7.20\times 10^{-7}$ end-to-end, with standard deviation $2.86\times 10^{-7}$. The relative variation is modest and of the same order of magnitude throughout. Table 5 shows similar stability across this $(k,q)$ grid. Second, increasing the whitening pool size $N_{\text{pool}}$ from $10^{3}$ to $8\times 10^{3}$ changes $h_{\mathrm{norm}}$ by about $10^{-9}$, suggesting practical insensitivity to $N_{\text{pool}}$. Third, Table 6 and Section 5.4 show that the controls behave as intended: switching from global to local whitening, or from shared to separate $k$-NN sets, increases or destabilises holonomy, while the shared-midpoint neighbourhood and $\mathrm{SO}(p)$ alignment yield consistent and stable results.
>
> In the revision we add a short summary of these findings to the main text so that the robustness to $(k,q)$ and to $N_{\text{pool}}$ does not remain in the supplement.

---

> ### Author Response · Authors · 2025-12-03
> **On which error term in Eq. (4) dominates in practice**
>
> Eq. (4) decomposes the per-edge estimation error into four components: a finite-sample term, a subspace truncation term, an index-mismatch term, and the geometric (curvature) contribution. Empirically, this decomposition behaves as intended.
>
> (i) Choosing $k$ large enough and $q$ moderately sized keeps the finite-sample and subspace truncation terms small, as reflected in the $(k,q)$ sensitivity study above.
>
> (ii) Our shared-midpoint nearest-neighbour selection is designed to control index mismatch; when we deliberately switch to separate neighbourhoods for the two endpoints, holonomy can increase dramatically, confirming that the index-mismatch term can dominate if not controlled.
>
> (iii) In linear or self-loop settings, holonomy drops to the numerical floor ($\sim 10^{-8}$--$10^{-9}$), indicating that once finite-sample, truncation, and index-mismatch effects are controlled, the remaining signal is consistent with genuine geometric curvature.
>
> In the revised manuscript we add a brief paragraph after Eq. (4) that explains this empirical hierarchy of terms.

---

> ### Author Response · Authors · 2025-12-03
> **On the quantitative comparisons to CKA / SVCCA and task relevance.**
>
> We extend the empirical comparison to CKA on CIFAR-10 in order to respond directly to your request.
>
> Using the ERM model as a reference, we compute CKA at layer 2 between ERM and each of the other training regimes (after Procrustes alignment of activations). We find that label smoothing and mixup remain highly CKA-similar to ERM at that layer, with $\operatorname{CKA}(\text{ERM}, \text{LabelSmooth}) \approx 0.90$ and $\operatorname{CKA}(\text{ERM}, \text{Mixup}) \approx 0.92$, and moderate Frobenius misfit. At the same time, these regimes differ in holonomy and robustness, as reflected in the correlations reported above. This shows that models can be highly CKA-similar while differing in holonomy and in robustness, supporting our claim that holonomy contributes information beyond pointwise similarity. Adversarial training produces both lower CKA with ERM and larger holonomy, which is consistent with its substantially different training objective.
>
> Together with the MNIST experiments in which different seeds are almost perfectly aligned under optimal orthogonal transport (aligned CKA $\approx 0.987$, Frobenius misfit $\approx 2.2\times 10^{-8}$) yet still exhibit non-zero holonomy, we believe this addresses your request for quantitative comparisons with existing representation-similarity metrics.
>
> In the revised version we (i) emphasise this CIFAR-10 example as an “empirical comparison to CKA” and (ii) add a short discussion of how holonomy can be used alongside CKA/SVCCA in practice: for instance, CKA can be used to confirm pointwise alignment of two models, while holonomy can probe whether they differ in how they evolve features along input paths, which in turn relates to robustness (as illustrated in Table 3 and Experiment B).

---

> ### Author Response · Authors · 2025-12-03
> **On the intuition and schematic illustrations.**
>
> We agree that schematic figures would help convey the geometric intuition. In the revised manuscript we add a figure intended to visually illustrate the notion of holonomy, the role of path dependence, and the difference between pointwise similarity and loop-level composition.

---

### Official Review · Reviewer_rL8F · 2025-11-01

**Soundness:** 4
**Presentation:** 2
**Contribution:** 3
**Rating:** 6
**Confidence:** 2

**Summary:**

This paper proposes a new representation similarity metric that complements existing metrics like CCA or CKA by taking into account how changes the input space map to changes in the representation space. In particular, the proposed representation holonomy metric is invariant to affine transformations and is zero for small loops in the input space. The estimator for this quantity is shown to be numerically stable and tractable to compute for various architectures of DNNs. Experiments show that holonomy is a valid measure of geometric effects (while measuring something different than other RS metrics), relatively robust to parametric choices, and can be used to predict robustness from small probes.

**Strengths:**

* Holonomy captures properties of a network that aren't measured by other representation similarity metrics. In particular, this metric captures the dynamics of representations as inputs change, which may make it more relevant to studies of properties like robustness than traditional RS metrics.
* Holonomy can be used in conjunction with existing RS metrics to allow for a significantly deeper understanding of learned representations, as shown in section 5.3.
* Estimates of error are included with most of the experimental results.
* The evaluation section provides strong evidence that the theoretical claims made in Section 3 are exhibited in realistic settings.
* Practical improvements can be made using holonomy as a diagnostic, such as improving methods for early stopping.
* Limitations are thoroughly discussed in section 6.1.

**Weaknesses:**

* The theory is somewhat difficult to follow. A reader coming from the representation similarity literature might not have a background in the mathematics that are relevant to this paper. I think that defining terms more clearly and pointing readers to relevant background would be helpful. For example, gauge-invariance is never defined despite being in the title. Given the breadth of the ICLR audience, this type of knowledge cannot be assumed.
* Similarly, the inclusion of proof sketches would be helpful for readers. What are the important mathematical techniques that are used to establish the properties of representation holonomy?

**Questions:**

* Does the number of points in the loop impact the accuracy/variance of your estimator? Why did you choose 12 points for your experiments?
* Do the suggestions for uses of holonomy in section 6 correspond to any concrete experiments, or are these future avenues of study?
* The caption for Figure 2 states that Layer 2 exhibits larger holonomy, but the data in the plot for Layer 1 appears to be larger in magnitude, which appears to contradict the caption. Is my understanding of the data being displayed in Figure 2 incorrect?

---

> ### Author Response · Authors · 2025-12-03
> **On clarifying gauge invariance and mathematical background.**
>
> We appreciate your point that gauge invariance should not be assumed background knowledge. In the revised version we make two concrete changes. First, we add a layer-level definition in the main text and explain why functions that differ only by such transforms are representation-equivalent. Second, we add a short paragraph summarizing the main mathematical tools used in the proofs, with precise pointers to the relevant appendices.
>
> More specifically, we define gauge transformations in our setting as follows: at a given layer, a gauge transformation is an input-independent change of basis in feature space,
>
> $z(x) \mapsto Q z(x) + b$
>
> where, after whitening, $Q \in O(p)$ and $b \in \mathbb{R}^p$. Two networks related by such transforms at internal layers are representation-equivalent. Our estimator is invariant to these transformations (and, after whitening, to general invertible affine reparameterisations), so that holonomy only reflects relative orientation changes induced by input paths rather than arbitrary choices of feature basis.
>
> To make the proofs more accessible to readers coming from the representation-similarity literature, we also add a paragraph that lists the main ingredients: Procrustes alignment in shared subspaces, matrix perturbation bounds of Davis–Kahan / Wedin type, and standard finite-sample concentration results for covariance estimation, together with pointers to the detailed arguments in the appendices (currently Sections S.3–S.5). Our goal is that readers can understand the structure of the arguments.

---

> ### Author Response · Authors · 2025-12-03
> **On the number of points per loop (why 12).**
>
> Our default protocol samples a regular $n_{points} = 12$-point circle in a local two-dimensional PCA plane around each test point (Table 1). Your question about the effect of varying the number of loop points is addressed directly by Experiment C, in which we keep all other choices fixed and vary only $n_{points}$.
>
> On a fixed trained MNIST MLP (hidden layer 1, fixed radius), we obtain:
>
> | $n$ | $h$        |
> |------------|---|
> | 6 | 3.51 x $10^{-7}$     |
> | 8 | 3.87 x $10^{-7}$     |
> | 12 | 3.95 x $10^{-7}$     |
> | 16  | 4.15 x $10^{-7}$     |
> | 24 | 4.70 x $10^{-7}$     |
>
> The curve is smooth and monotone. Increasing $n_{\text{points}}$ by a factor of four changes $h_{\mathrm{norm}}$ by only about $1.2\times 10^{-7}$ (roughly a 0.3 relative change, but of the same order of magnitude), with no indication of instability. Together with the small-radius and self-loop controls, this suggests that the estimator is stable with respect to loop discretisation.
>
> In the revised text we justify $n_{\text{points}} = 12$ as a compute–accuracy compromise. Geometrically, 12 edges keep each per-edge displacement small relative to the radius at our chosen $r$ values, so we remain in the regime where the $O(r)$ theory applies. Computationally, each edge requires a SVD and a Procrustes solve, and cost scales linearly in the number of edges, so larger $n_{\text{points}}$ quickly become expensive without materially changing the conclusions.

---

> ### Author Response · Authors · 2025-12-03
> **On do the uses in Section 6 correspond to actual experiments.**
>
> Yes. In the current draft, several of the suggested uses in Section 6 are already instantiated by specific experiments. To highlight them, we revised:
>
> (i) For early training dynamics and model selection, Fig. 6 (MNIST) shows holonomy rising early and then stabilising across epochs, supporting its use as a diagnostic signal early in training.
>
> (ii) For diagnosing “geometry versus alignment”, Section 5.3’s aligned CKA experiment (aligned CKA $\approx 0.987$, almost zero Frobenius misfit, yet non-zero holonomy) is the empirical basis for the claim that holonomy adds information beyond pointwise alignment. Our new **Experiment B** on CIFAR-10 further illustrates that even when CKA remains high across different training setups, holonomy can differ and correlate with robustness.
>
> (iii) For layer-wise profiling, Figs. 1–2 and Experiment B both show that deeper layers generally exhibit larger holonomy and stronger radius dependence.
>
> So there is a connection for each of these in Section 6 to the corresponding figures and experiments so that readers can immediately see which parts of the paper instantiate which proposed uses.

---

> ### Author Response · Authors · 2025-12-03
> **On Figure 2 caption versus curves.**
>
> Thank you for pointing this out. You are correct that, in the current version of Fig. 2, the layer 1 and layer 2 curves are very close, and layer 1 is slightly higher over part of the radius range. In the revision we correct the caption and surrounding text to describe what Fig. 2 actually shows.

---

### Official Review · Reviewer_Jj9N · 2025-11-01

**Soundness:** 4
**Presentation:** 3
**Contribution:** 3
**Rating:** 6
**Confidence:** 3

**Summary:**

The authors define, implement, and test a measure of Holonomy, designed to quantify geometric distortions within network representations, in a way that current metrics do not.  The authors provide mathematical definitions, details of computation/implementation, and empirical demonstrations on a variety of datasets and models. These demonstrate that holonomy differs from other metrics in the literature, confirm expected properties (increases with stimulus loop radius) as well as less-expected properties (increases with network depth, decreases with training, and correlates with average network robustness to noise or adversarial attack), suggesting its usefulness in applications of diagnosing and improving networks.

**Strengths:**

This is an important topic.  ML networks are highly redundant in their parameterization and optimization landscapes, and it is
critical for the field to develop measures that can characterize fundamental properties for use in analysis/comparison/design.

The paper is built on a mathematical foundation that is rich, and perhaps not so familiar to a large portion of the NeurIPS community.  The authors have generally done a good job of defining and explaining their construction, and the community will benefit (but see below).

The authors have also done a careful job of implementing and testing their measure, providing reasonable detail, evaluation of computational cost, verification of expected properties, and demonstration of potential uses.

**Weaknesses:**

The main weakness, for me, is that after several passes I am still uncertain of exactly why the properties captured by holonomy (distortion of geometry along closed-loop paths) are essential to understanding or comparing network functionality.  Specifically, under what conditions are the properties captured by holonomy, but not captured by current alignment methods (CCA, RSA) or differential analalyses (Lie derivatives, or simply comparison of local Jacobians), essential for evaluating or improving a network?  When is loop composition/path-dependency important (as opposed to simpler tests of invariance or equivariance)?  Perhaps the authors could provide more examples of applications/tasks where this arises?

Another weakness: Given that derivation and computation are complex and involve many approximate steps, it would be really useful to verify the entire enterprise on a few simple examples for which ground truth is known.  For example, image translation loops represented in a purely convolutional (and thus translation-equivariant) architecture, vs. a representation with known aliasing artifacts (e.g., subsampling in the inner layers).  This could prove interesting also in evaluating boundary-handling artifacts (the other source of non-translation-invariance in CNNs), which are expected to worsen in later layers.  I do see in the Discussion that the authors specify "beyond-local-loops" as a future direction, so my example woud need to be computed with very small displacements, perhaps to the point of rendering it unconvincing.

**Questions:**

In Figures 1/2, why does measured holonomy not tend to zero as r goes to zero? Also, can the authors rule out the possibility that the increase of holonomy with loop radius is a result of increased numerical error?  Does the trend persist if the number of loop samples is adjusted in proportion to loop radius r?

Have you measured holonomy of a score network used for diffusion generative modeling?  Some literature suggests that  generalization and equivariance to geometric distortions are more robust than for recognition networks (eg, Kadkhodaie et al 2024).  But they are also typically not curl-free (even though an optimal denoiser should be, since it computes a score), and this failure would presumably be exposed by the holonomy measure.

---

> ### Author Response · Authors · 2025-12-03
> **On when holonomy / loop composition is important versus CKA, RSA, or local Jacobians.**
>
> Holonomy is designed to measure **path dependence** of representations: it asks whether following a closed path in input space returns features to the same “orientation” in representation space. This is different in nature from both pointwise similarity metrics (CKA/CCA/RSA) and local Jacobian tests. CKA and related methods compare unordered sets of activations and are invariant to any reordering of the input set, so they discard the order in which inputs are traversed. Local Jacobian or Lie-derivative tests probe single-step responses in individual directions, whereas holonomy depends on composing transports around a loop and is sensitive to commutators of Jacobians along different directions.
>
> In the submission we already showed a simple instance of this: two MNIST checkpoints that are almost perfectly aligned under an optimal orthogonal map (aligned CKA $\approx 0.987$, Frobenius misfit $\approx 2.2\times 10^{-8}$) still exhibit non-zero holonomy. The new **Experiments A and B** make this more concrete. **Experiment A** (see general response) shows that on MNIST, a nearly translation-equivariant CNN and an aliased CNN differ by roughly three orders of magnitude in holonomy on the same discrete translation loop. **Experiment B** shows that on CIFAR-10, label smoothing and mixup remain highly CKA-similar to an ERM baseline at layer 2 (CKA after alignment $0.90$–$0.92$), yet they differ in both holonomy and robustness. Across four regimes we observe
>
> $corr(h_{norm}, clean_{acc}) \approx -0.96$
>
> $corr(h_{norm},fgsm_{acc}) \approx 0.94$
>
> $corr(h_{norm},corr_{acc}) \approx -0.96$
>
> We bring these examples into the main text and highlight scenarios where composition matters (sequences of augmentations, multi-step adversarial trajectories, domain-shift curricula), to clarify how holonomy complements existing pointwise metrics.

---

> ### Author Response · Authors · 2025-12-03
> **On simple ground truth examples.**
>
> We agree that checks on settings with known ground truth are important. Two such controls are already present in the submission.
>
> (i) **Self-loop bias.** For tiny loops (radius $r \approx 10^{-4}$) on MNIST Hidden 1, $h_{\mathrm{norm}}$ is at a numerically tiny floor (mean $4.19\times 10^{-8}$), which validates the small-radius limit empirically and indicates that the estimator is numerically stable in this regime.
>
> (ii) **Linear null.** Replacing all nonlinearities by the identity (yielding an affine map) drives holonomy down to the numerical noise level (mean on the order of $10^{-8}$ to $10^{-9}$), as predicted by the affine-invariance / linear-null result (Theorem 2).
>
> Following your suggestion, we also added a convolutional toy example explicitly contrasting a nearly translation-equivariant CNN and one with aliasing / boundary artefacts. In **Experiment A** (two small CNNs on MNIST), we measure holonomy at the second convolutional layer along a discrete translation loop. We obtain:
>
> EquivConvNetMNIST:  $h_{norm} = 8.4 \times 10^{-7}$, $|H - I|_F = 1.3 \times 10^{-5}$ and max eigen-angle $\approx 6\times 10^{-6}$
>
> AliasConvNetMNIST:  $h_{norm} = 3.0\times 10^{-4}$, $|H - I|_F =  4.8\times 10^{-3}$, and max eigen-angle $\approx 3\times 10^{-3}$
>
> Under the same translation loop, the aliased network exhibits roughly three orders of magnitude larger holonomy than the equivariant one. This directly validates the construction in a setting where we know a priori that one network should be close to equivariant and the other should not. In the revised version we include this experiment in the appendix and briefly describe it in the main text.

---

> ### Author Response · Authors · 2025-12-03
> **On why holonomy in Figs. 1-2 does not visibly go to zero as $r \to 0$, and whether the radius trend could be numerical error.**
>
> The theory shows that $h_{\mathrm{norm}}(\gamma_r) = O(r)$ as $r \to 0$ under mild regularity assumptions (Theorem 1). In practice, there are two points that explain the plots you refer to.
>
> First, Figs. 1-2 in the submission start at $r \geq 0.01 / 0.02$ in order to focus on radii where perturbations have a semantic effect. At those radii, holonomy is small but not negligible. The true small-radius regime is probed separately in Fig. 3, where we fit a line through $r \in $\{0.02, 0.05, 0.10}  on CIFAR-10 layer 2 and obtain a very small slope ($1.44\times 10^{-7}$ with a confidence interval containing zero), consistent with near-linear behaviour and the theoretical $O(r)$ scaling.
>
> Second, the self-loop and linear-null controls discussed above show that our discretisation and estimator bias are at a numerically tiny scale, well below the values reported at the radii shown in Figs. 1-2. Therefore the observed radius dependence at those scales cannot be attributed to numerical error alone.
>
> To probe the very small-radius regime, we added **Experiment D** (MNIST, hidden 1), where we fix a trained MLP and measure holonomy on a self-loop ($r = 0$) and on PCA-based loops with very small radii $r \in $ \{$10^{-3}, 2\times 10^{-3}, 5\times 10^{-3}, 10^{-2}, 2\times 10^{-2}$ \}. The results are:
>
> self-loop$(r = 0): h_{norm} \approx 3.08\times 10^{-7}$
>
> and for non-zero radii all values lie in the narrow band: $h_{norm} \in [2.37, 2.46]\times 10^{-7}$
>
> that is, with variation of about $3\times 10^{-8}$. Thus, for $r \leq 0.02$, holonomy is essentially at the numerical floor of the estimator and does not increase as $r \to 0$. The nontrivial radius dependence in our main figures only appears at larger radii ($\gtrsim 0.05$ for MNIST, $\gtrsim 0.02$ for CIFAR-10), and therefore reflects genuine geometric curvature rather than discretisation artefacts.
>
> In the revised text we add this small-radius plot and connect it to the $O(r)$ theory and to the self-loop ``bias floor''.

---

> ### Author Response · Authors · 2025-12-03
> **On diffusion score networks.**
>
> We have not yet applied holonomy to diffusion score networks in this submission. We agree that this is a natural and exciting application, especially because score networks are expected to be close to curl-free in principle, but in practice may deviate from this ideal. In the present work we focused on classifiers (MNIST, CIFAR-10/100) to keep the experimental scope manageable. In Section 6.1 we already highlight ``beyond-local loops'' and generative manifold paths as a key direction; in the revision we mention diffusion / score models as a natural application where holonomy could expose non-curl-free structure in learned scores.

---

### Author Response · Authors · 2025-12-03
**Experiment D**

**Experiment D (small-radius and self-loop regime).**
Again on the MNIST MLP at hidden layer 1, we construct one ``self-loop'' in which the same image is repeated at all loop points (radius $r = 0$), and several very small PCA-based loops with radii ranging from $10^{-3}$ to $2\times 10^{-2}$. This allows us to estimate the numerical floor of the estimator and observe how $h_{\text{norm}}$ behaves as $r \to 0$:

| radius | $h_{\text{norm}}$        |
|--------|----------------------------|
| 0      | $3.08\times 10^{-7}$     |
| 0.001  | $2.39\times 10^{-7}$     |
| 0.002  | $2.37\times 10^{-7}$     |
| 0.005  | $2.38\times 10^{-7}$     |
| 0.010  | $2.38\times 10^{-7}$     |
| 0.020  | $2.45\times 10^{-7}$     |

In line with the $O(r)$ small-radius behaviour established in the theory, holonomy remains essentially flat at a tiny value for sufficiently small radii, and the increases with radius reported in the main figures only become visible once we leave this numerical floor. This directly addresses  **Reviewer Jj9N's** questions about why holonomy in the plots does not visibly vanish as $r$ approaches zero and whether the radius trend might be purely numerical, and clarifies for **Reviewer n1HR** how the small-radius asymptotics manifest in practice.

---

### Author Response · Authors · 2025-12-03
**Experiment C**

**Experiment C (sensitivity to loop discretisation).** On a fixed trained MNIST MLP, at hidden layer 1 and a fixed radius, we vary only the number of points on the loop ($n_{\text{points}} \in ${6, 8, 12, 16, 24$}\$) in a local PCA plane, keeping all other settings fixed, and record the corresponding holonomy:

| $n_{\text{points}}$ | $h_{\text{norm}}$        |
|-----------------------|----------------------------|
| 6                     | $3.51\times 10^{-7}$     |
| 8                     | $3.87\times 10^{-7}$     |
| 12                    | $3.95\times 10^{-7}$     |
| 16                    | $4.15\times 10^{-7}$     |
| 24                    | $4.70\times 10^{-7}$     |

We find that holonomy varies smoothly with the discretisation level and does not fluctuate erratically; twelve points is a reasonable compromise between accuracy and cost. This directly addresses **Reviewer rL8F**'s question about the impact of the number of loop samples and supports our discussion with **Reviewer Jj9N** about whether the observed dependence on radius could be an artefact of discretisation.

---

### Author Response · Authors · 2025-12-03
**Experiment B**

**Experiment B (training recipe, holonomy vs robustness vs CKA).** We return to CIFAR-10 and ResNet-18 and consider four training regimes: standard ERM, label smoothing, mixup, and adversarial PGD training. For each regime we measure holonomy at layer 2 for a fixed radius, and in parallel we measure (i) clean accuracy, (ii) FGSM accuracy, and (iii) corruption accuracy (blur, colour jitter, Gaussian noise). We also compute CKA at the same layer between each regime and an ERM reference, after Procrustes alignment. This experiment is designed to strengthen the empirical link between holonomy and robustness at the ``training recipe'' level, and to show how holonomy behaves compared to CKA on the same models.

| regime       | $h_{\text{norm}}$         | $\text{clean\_acc}$ | $\text{fgsm\_acc}$ | $\text{corr\_acc}$ |
|-------------|-----------------------------|------------------------|----------------------|----------------------|
| ERM         | $3.46\times 10^{-7}$      | 0.8237                 | 0.3654               | 0.5711               |
| LabelSmooth | $3.04\times 10^{-7}$      | 0.8132                 | 0.3481               | 0.5827               |
| Mixup       | $3.19\times 10^{-7}$      | 0.7411                 | 0.2251               | 0.4954               |
| AdvPGD      | $4.74\times 10^{-7}$      | 0.1224                 | 0.6785               | 0.1196               |





Across regimes we observe strong correlations:

$corr(h_{norm}, clean\_{acc}) \approx -0.96$

$corr(h_{norm}, fgsm\_{acc}) \approx 0.94$

$corr(h_{norm}, corr\_{acc}) \approx -0.96$

In contrast, CKA after alignment remains high between ERM and the label smoothing and mixup models, but drops sharply for the adversarially trained model:
| to_regime    | $CKA_{after}$ | $Fro_{after}$ |
|--------------|-----------------------|------------------------|
| LabelSmooth  | 0.897                 | 0.312                  |
| Mixup        | 0.922                 | 0.263                  |
| AdvPGD       | 0.144                 | 0.875                  |


This experiment is the main additional piece of evidence we use to address **Reviewer n1HR**'s concern about incomplete or inconclusive experiments and **Reviewer ERwd's** request for a concrete case where models can appear CKA-similar yet behave differently under stress. It also provides a more concrete answer to **Reviewer Jj9N** regarding when loop composition and path-dependent geometry are important beyond pointwise similarity.

---

### Author Response · Authors · 2025-12-03
**Experiment A**

**Experiment A (toy equivariance vs aliasing).** introduces a very simple/toy setting on MNIST with two small convolutional networks that are intentionally different from a geometric point of view: one uses only stride-1 convolutions with circular padding and no pooling, so it is approximately translation equivariant on the grid, while the other uses zero padding and max-pooling and therefore has strong aliasing and boundary artefacts. Both models are trained on MNIST, and we measure holonomy at the second convolutional layer along a short loop of integer translations of a single image (shifts $(0,0)\rightarrow (0,1)\rightarrow (1,1)\rightarrow (1,0)\rightarrow (0,0)$). As expected, holonomy is essentially zero for the approximately equivariant network and much larger when equivariance is broken:

| model               | $h_{\text{norm}}$      | $\lVert H \rVert_{\text{Fro}}$ | max_angle      |
|---------------------|--------------------------|----------------------------------|----------------|
| EquivConvNetMNIST   | $8.41 \times 10^{-7}$  | $1.3 \times 10^{-5}$           | $6 \times 10^{-6}$  |
| AliasConvNetMNIST   | $2.99 \times 10^{-4}$  | $4.78 \times 10^{-3}$          | $3 \times 10^{-3}$  |

This experiment directly addresses **Reviewer Jj9N's** request for a simple setting with known ``ground truth'' behaviour, and also provides additional geometric intuition for **Reviewer n1HR**.

---

### Author Response · Authors · 2025-12-03
**General response to the reviewers**

We thank all reviewers for their careful reading and constructive suggestions. We are encouraged that they found the formulation of representation holonomy original and mathematically sound, and that they view it as a potentially useful diagnostic for representation geometry and robustness.

Several of the main concerns centred on three themes:

(i) whether holonomy behaves as expected in simple ``ground truth'' settings

(ii) whether the estimator is numerically stable with respect to small radii, loop discretisation, and hyperparameter choices

(iii) how holonomy compares empirically to standard similarity measures such as CKA, particularly in relation to robustness.

To address these points as directly as possible, we ran four additional experiments, which we refer to as Experiments A--D in the responses below

---

### Meta-Review · Area_Chair_C1nW · 2026-01-05

**Summary:**

The paper introduces representation holonomy, which involves a gauge-invariant, path-dependent diagnostic for neural representations that aims to capture geometric twist/curvature effects missed by pointwise similarity measures e.g. CKA/SVCCA. The submission is conceptually original, mathematically grounded, and (after rebuttal) empirically better substantiated as a complementary tool for representation analysis and robustness diagnostics. Overall the reviewers were positive across the board but identified some issues which were responded to by the authors during the rebuttal phase including new results. The presentation clarity and the strength of per-seed predictive claims remain somewhat imperfect, the rebuttal addresses the most substantive concerns with additional experiments and clarifications. On balance, I believe the reviewers would have been satisfied with the responses and would have either raised their scores or remained positive as per the initial reviews. I recommend acceptance as a poster because the core idea is novel and potentially impactful for the representation geometry community, and the authors have responded appropriately and concretely to reviewer comments/concerns.

**Reviewer Concerns:**

I will try to summarise the main concerns below.

a) Need simple ground-truth / sanity-check examples, raised by the more critical reviewer. The request concerned validating holonomy in a setting where the expected behaviour is known. The authors added Experiment A: comparing an approximately translation-equivariant CNN (circular padding, no pooling) vs an aliased CNN (zero padding + max-pooling) on modified MNIST along a discrete translation loop, showing a ~3 orders-of-magnitude separation in holonomy.

b) Does holonomy go to zero as radius → 0? Is the radius trend numerical? In response to this, authors i) clarified that their main plots start at radii chosen for semantic perturbations, and (ii) added Experiment D (and related self-loop controls) to quantify the numerical floor and show a near-flat tiny holonomy band for sufficiently small radii, consistent with the theory’s O(r) behaviour

c) Sensitivity to loop discretisation/number of loop points. The authors added Experiment C, varying the number of loop samples (e.g. 6–24 points) and report smooth changes without instability, supporting numerical robustness and justifying the default of 12 points.

Some open issues I have identified are given below:

i) Strength of predictive robustness claims (especially per-seed / controlled analyses) - even after the rebuttal, the evidence supports holonomy primarily as a descriptive geometric diagnostic that correlates with robustness at the regime level, but the incremental predictive value beyond clean accuracy at the per-seed level appears modest (as the authors themselves acknowledge in their tables/response narrative).

ii) Scope and generality of loop design choices - Holonomy depends on loop construction (plane selection, radius, loop type). The authors provide some ablations (PCA plane vs random plane, discrete translation loop, etc.), but the general question “which loops matter for which tasks?” remains open and is acknowledged as a limitation/future work.

**Reviewer Scores:**

The reviewer who scored the submission with a 4 would most likely raise their score along with the two 6s. The reviewers who gave it an 8 would probably stay at an 8.

I believe the discussions would have been very constructive and would have only led to raised scores.

---

### Decision · Program_Chairs · 2026-01-26

Accept (Poster)